# Review of a 25-Year Experience in the Management of Ovarian Masses in Neonates, Children and Adolescents: From Laparoscopy to Robotics and Indocyanine Green Fluorescence Technology

**DOI:** 10.3390/children9081219

**Published:** 2022-08-12

**Authors:** Esposito Ciro, Coppola Vincenzo, Cerulo Mariapina, Del Conte Fulvia, Bagnara Vincenzo, Esposito Giorgia, Carulli Roberto, Benedetta Lepore, Marco Castagnetti, Gianluigi Califano, Maria Escolino

**Affiliations:** 1Pediatric Surgery Unit, Federico II University of Naples, 80131 Naples, Italy; 2Pediatric Surgery Unit, Polyclinic G.B. Morgagni, 95125 Catania, Italy; 3Pediatric Urology Unit, Pediatric Hospital Bambin-Gesù, 00165 Rome, Italy

**Keywords:** ovarian masses, children, laparoscopy, robotics, ICG fluorescence technology

## Abstract

Background: Ovarian masses in pediatric populations are the most common abdominal masses in young girls. In neonates, the majority of masses are benign while in children and teen-agers the risk of malignancy exists. The aim of this study is to perform a 25-year experience retrospective analysis of clinical and therapeutic aspects of ovarian tumors in girls, in order to show how the development of minimally invasive technology has changed the management of this pathology. Methods: The records of patients under the age of 18 who were operated in three pediatric surgical units due to ovarian mass, in the last 25 years, were reviewed retrospectively. The study group comprised 147 patients operated between 1996 and 2021 with a diagnosis of ovarian masses. Data involved were demographical, surgical, follow-up and final diagnosis. We analyzed the type of surgical technique, intra-operative data (operative time, the use of different technologies), complications, length of stay and long-term follow-up. Based on these data, we assessed how the surgical approach to ovarian masses has changed in the last 25 years in newborns and young girls. Results: The patients ages ranged between 7 days and 15 years (median, 59 days). All the procedures were completed in laparoscopy or robotics without conversion in open surgery. One-hundred and eleven patients were neonates; they all had follicular cysts and they were all managed in laparoscopy using 1 or 3 trocars. In 80/111 patients (72%), a small part of ovarian parenchyma was saved; in 31/111 patients (28%), in which the ovarian parenchyma was not available, an ovariectomy was performed. Patients in which we saved a small part of ovary, at long term follow-up (minimum follow-up of 12 years) (29/80, 36%), developed a normal ovary at US control. Thirty-six were older patients. They had a histological diagnosis of benign (30) or malign (6) tumors. All the patients (8/36) with a pre-operative suspicion of ovarian malignancy received an ovariectomy and an adnexectomy using sealing devices. In the last 10 years in all the children, except neonates, we adopted sealing devices and, in the last 4 years, in 20 cases, we always adopted ICG fluorescence technology to check ovarian vascularization in case of torsion or to check lympho-nodes condition in case of malignancy. Conclusions: In neonatal ovarian cysts, surgical management remained unchanged and an ovarian sparing procedure is always indicated and the long-term follow-ups confirm this hypothesis. The principal innovation in this age period is the use of ICG fluorescence technology to check ovarian vascularization in case of torsion. In teenagers, the decision-making strategy is based on the tumoral markers and on the morphological aspects of the mass. Robotics cystectomy or ovariectomy now-days represents the safer and faster way to perform this. Sealing devices are essential tools for dissection and resection to avoid bleeding. ICG fluorescence technology in all ages is fundamental to check ovary vascularization after detorsion or to check lympho-node status in case of malignancy. All the suspected lesions have to be removed with an endo-bag.

## 1. Introduction

The most common abdominal masses in pediatric populations are ovarian masses. Ovarian cysts are the most frequent cause of an abdominal mass in the fetus and in newborns [1], occasionally are neoplasm and almost never malignancies. They can produce a variety of striking clinical findings clearly described prior to the sonographic era [2,3,4]. Usually, they are functional cysts and often they do not require surgery. Even in children and teenagers, the most frequent ovarian masses are cysts, although the risk of malignancy exists. Why a particular follicle becomes a large cyst is not known, but fetal follicles are sensitive to hormonal influences [5]. Any ovarian or adnexal mass could cause ovarian torsion, which can occur secondary to excessive mobility of the ovary or fallopian tube. In either situation, torsion is a high-stakes imaging diagnosis and surgical emergency. [6] For these reasons, early diagnosis is necessary to reduce the risk of complications and to improve the prognosis for children with malignant neoplasms. Even though controversy exists concerning the optimal management of the asymptomatic cyst, the management is based generally on its size and whether the cyst is sonographically simple or complex [7]. Usually, a simple cyst smaller than 2 cm in diameter should be left alone and can be reassessed with sonographic examinations [8]; in fact, in 65% of cases, there could be a spontaneous resolution in 4–6 months [9]. Bigger size, complex or symptomatic cysts warrant intervention. Although ovarian salvage is touted widely for ovarian cysts and aspiration alone or partial cystectomy is satisfactory, a total ovariectomy has been performed increasingly often. The reason for this is due to the difficult identification of the mass in the context of the ovary. Especially in case of ovarian cysts, the use of more conservative surgery appears necessary to avoid an ovariectomy. ICG (Indocyanine Green) enhanced fluorescence was proved to be very useful to improve intra-operative anatomic view during laparoscopic and robotic removal of ovarian masses, especially to check vascularization in case of torsion. In case of neonatal cysts, the sparing procedure is always indicated. On the other hand, in case of cysts involving children and teenagers, minimally invasive surgery has improved a lot in recent years. The aim of this study is to perform a 25-year retrospective analysis of clinical and therapeutic approaches of ovarian tumors to show how minimally invasive technology has changed the management of this pathology.

## 2. Materials and Methods

We retrospectively recorded the data of patients undergoing surgery for ovarian mass before 18 years of age during the last 25 years in three pediatric surgical units. The study group comprised 147 patients operated for ovarian tumor in between 1996 and 2021. We divided patients into two groups based on their age. The first group (G1) consisted of 111 newborns between 1 and 6 months of age; the second group (G2) consisted of 36 children between 6 months and 18 years of age (Table 1).

In our study, we involved demographical and surgical data, follow-up and final diagnosis. We analyzed the type of surgical technique, intra-operative data (operative time, the use of different technologies), complications, length of stay and long-term follow-up. In the teenager group, tumors markers were always analyzed before surgery (CA125, CA19-9, CEA, AFP). Beta-hcg was performed in all teenagers to rule out pregnancy. Based on these data, we assessed how the surgical approach to ovarian masses has changed in the last 25 years in newborns and young girls. We focused our attention on the new system based on ICG technology, which was adopted in recent years in 10 laparoscopic and in every robotic procedure of ovarian mass removal, for a total of 20 cases (Table 2). ICG solution was always intra-operatively injected intra-venously. In these cases, we tried to understand if the ICG technology was useful to the surgeon in terms of decision-making and therapeutic purposes.

Statistical analysis was carried out using the Statistical Package for Social Sciences (SPSS Inc., Chicago, IL, USA), version 13.0. The categorical variables were compared using **χ**^2^ tests. Significance was defined as *p* < 0.05.

The study was reviewed and approved by ethics committee.

## 3. Results

All patients were managed with mini-invasive surgery: in G1, 1-trocar or 3-trocar laparoscopy was used; in G2, 3-trocar laparoscopy (28 cases) or robotic surgery (8 cases) was used. In 91 cases, the right side was involved; in 56 cases, it was the left side. The patients ages ranged between 7 days and 15 years (median, 59 days).

All the procedures were completed in laparoscopy or robotics without conversion in open surgery. 

One-hundred and eleven patients (G1) were neonates; they had all follicular cyst and they were all managed in laparoscopy using 1 or 3 trocars. In G1, a small part of ovarian parenchyma was saved in 80 patients (72%), while in 31 patients (28%) the ovarian parenchyma was not available, so they received an ovariectomy (*p* = 0,0001) (Figure 1). No cases of malignancy were recorded in the newborn group from the histopathological analysis. 

In the last six cases of G1, ICG technology was adopted, and it allowed a better visualization of the mass compared to the surrounding tissue. With this technology it was possible to check ovarian vascularization in case of torsion. Patients in which we saved a small part of the ovary, at long term follow-up (minimum follow-up of 12 years) (29/80, 36%), developed a normal ovary at US control.

Thirty-six were older patients (G2). We managed 28 cases in laparoscopy and 8 cases in robotic-assisted surgery (in the last six years). They had a histological diagnosis of benign (30) or malign (6) tumors. In G2, positive tumor markers were noted in 3/30 patients (10%) with a benign lesion and in 4/6 (66%) with ovarian malignancy (χ^2^
*p* = 0,0014). All the patients with a diagnostic suspicion of ovarian malignancy underwent ovariectomy and adnexectomy using sealing devices (Figure 2). In 36 teenagers, the ovary was excised, placed inside an endo-bag, and extracted through the umbilicus. 

Of the 36 cases treated in G2, six of them were malignant (16.6%) and the pathological diagnosis was germ cell tumor (five cases) and sex cord tumor (one case). Ovarian torsion occurred in 13/37 (36%) and only in two of them intermittent abdominal pain presented as the primary symptom. 

Even in the G2 group, the ICG proved useful to check ovarian vascularization after ovarian detorsion; it allowed the evaluation of lymph-nodes condition in case of malignancy, for which oophorectomy and adnexectomy was necessary (Figure 3). The ICG system was used in the last 14 patients treated and it emerged that only in 3/14 (21.5%) a total ovariectomy was performed and it was possible to save the ovary (χ^2^ 3.82, *p* = 0.0494).

In G2, the mean operative time in patients treated before the introduction of ICG was 63.5 min compared to the mean operative time in patients treated with ICG system which was 39.2 min (t 6.8783, *p* = 0.00005). Operative time reduction with the use of ICG technology in newborns was not significant (*p* > 0,05) (Table 2).

In G1 there were no intra- and post-operative complications. In G2, one patient had an intra-operative massive bleeding that was controlled successfully with the sealing device. A total of 2/36 patients in G2, operated more than 10 years ago, had an incorrect pre-operative diagnosis of benignity. 

All patients in G2 underwent a follow-up, with outpatient visits and ultrasound scans every year without complications. The average hospital stay was 3 days for G1 and 2 days for G2. 

## 4. Discussion

Ovarian tumors in children and adolescents are a subject of interest for pediatric surgery centers, as most patients from these age groups require surgical management. 

Our 25-year retrospective study included all newborns and teenagers with ovarian lesion referring to three pediatric surgery centers in Italy. 

In the case of neonatal cysts, an ovarian sparing procedure is always indicated, as described in the literature [8,10,11]. In our series, in most cases it was possible to remove the mass and spare the ovary, with excellent long-term results. The procedure involves the removal of the ovary through the navel (Figure 4) and, only if healthy tissue is not available, an ovariectomy is indicated [12].

The surgical approach in case of children and teenagers should be different and a correct strategy is based on various factors, including the size of the mass, the morphological aspect and the presence or absence of positivity for tumor markers [13].

In our experience, tumor markers can be useful in evaluating ovarian masses; however, in 10% of our cases we had false positivity, so they could only assist in the diagnosis. Therefore, a new systematic method with high sensitivity and specificity for early diagnosis of ovarian cancer and new tumor markers needs to be identified [14].

Surgical excision is always indicated in the case of large dimension of the mass, doubts about the nature of the mass or risk of ovarian torsion. The best surgical approach is represented by minimally invasive surgery, laparoscopy, or, in recent years, robotics. The use of sealing devices is essential to perform dissection, avoiding bleeding, avoiding the use of metal clips on the vessels, thus making an easier procedure for surgeons.

In the case of ovarian masses in teenagers, the initial approach should always be conservative [15,16].

In a work of Hermans et al., it emerged that, in a population of 111 patients with ovarian masses, 28 patients (25.2%) were malignant. An important fact that emerged, however, was that in 46.4% of benign masses was performed a total ovariectomy [17]. This is due to various factors, mainly the fact that the surgeon is not sure of the benign nature of the mass and, more importantly, is unable to distinguish the mass from healthy tissue. 

This is not in line with the literature, which, on the contrary, supports the need to preserve the ovary during surgery. The incidence of malignant mass in children is very low [1,17,18]; for this reason, if no sign (i.e., palpable mass) or symptom (i.e., abdominal pain) suggest a malignancy, ovarian preserving surgery should be always considered as the first option to preserve fertility [19]. 

In recent years, we applied a new visualization system using ICG technology for minimally invasive surgery of ovarian masses. In some cases, this system allows for a better visualization of the margins of excision of the mass, and it can play a role in the intraoperative evaluation of a lesion, to have a correct decision-making process. In our study, in the last 14 cases treated with ICG, in fact, it was more possible to spare the healthy ovary and avoid ovariectomy, with a reduction in the number of total ovariectomy. Furthermore, the ICG system was essential to check the vascularization of the ovary following a torsion and evaluate if there was any ischemic damage (Figure 5). There appears to be overwhelming evidence supporting ovarian detorsion rather than oophorectomy for the management of ovarian torsion in pediatric patients [20]. Often, however, the surgeon has difficulty in understanding whether or not there has been ischemic damage to the tissue. The ICG proves to be very useful in this regard. 

According to our data, it seems also that the ICG system reduces operating time; however, further cases are necessary to confirm this data. 

The main limitation of the ICG system is the specific equipment needed in laparoscopy, which is not available in all centers with specific software for NIRF and which has high costs. For this reason, this system is useful for those who already have the ICG technology, especially for those who treat urological pathologies [21].

Concerning the removal of the ovarian mass in teenagers, we believe that the lesions should be removed with an endo-bag, which represents the safest and simplest way to avoid possible dissemination of the mass. All masses must be sent for pathological anatomy analysis. 

In conclusion, we think that, in recent years, all ovarian masses should be approached in mini-invasive surgery; robotic surgery is a new surgical-approach in ovarian masses in teenagers; the ICG system can be useful to evaluate the vascularization of the ovary in case of torsion and to make a correct decision during the surgery.

## Figures and Tables

**Figure 1 children-09-01219-f001:**
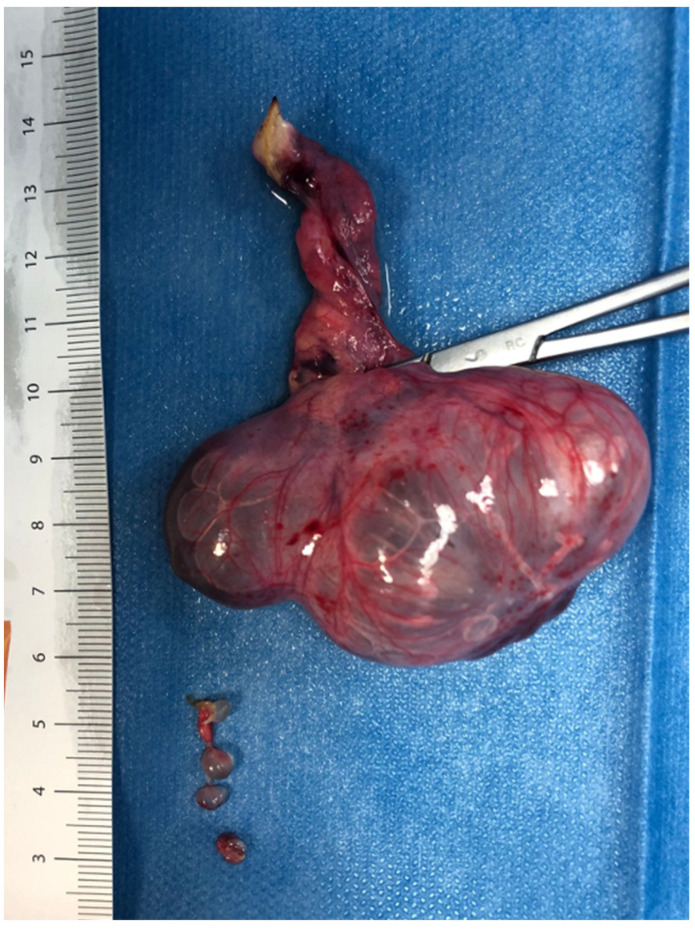
Laparoscopic ovariectomy for ovarian torsion.

**Figure 2 children-09-01219-f002:**
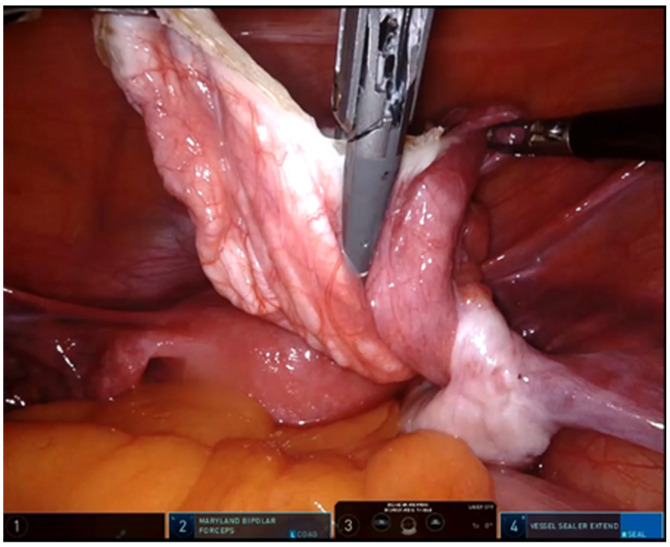
Use of sealing device in robotic removal of ovarian mass.

**Figure 3 children-09-01219-f003:**
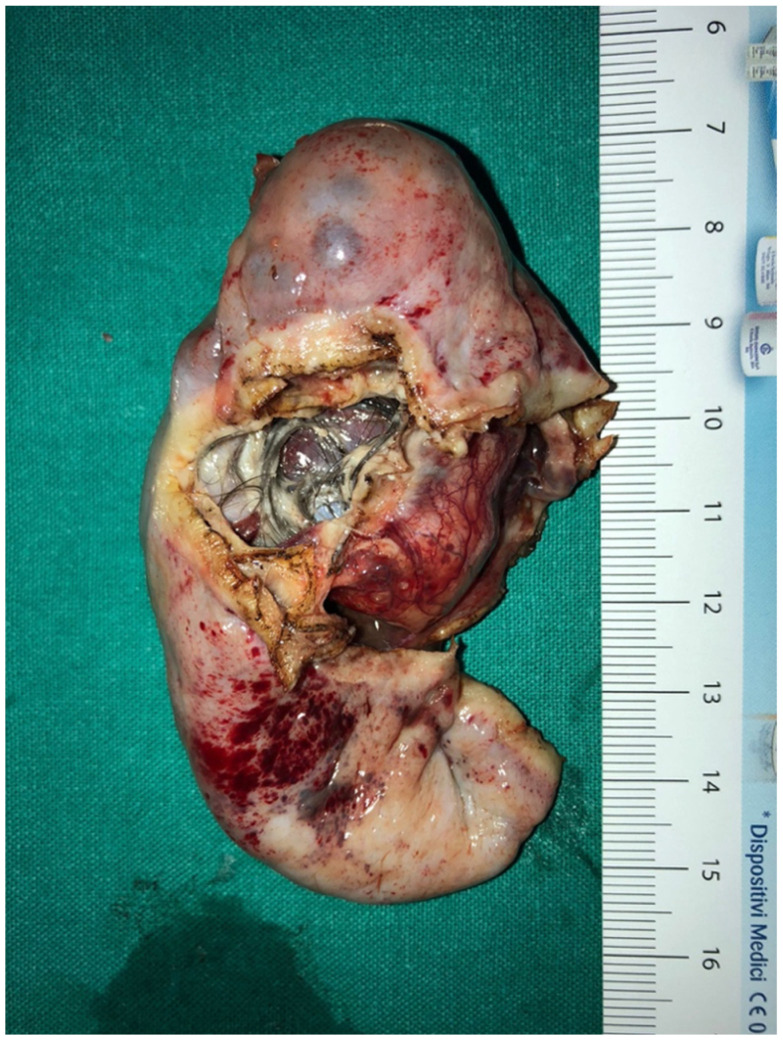
Immature Teratoma.

**Figure 4 children-09-01219-f004:**
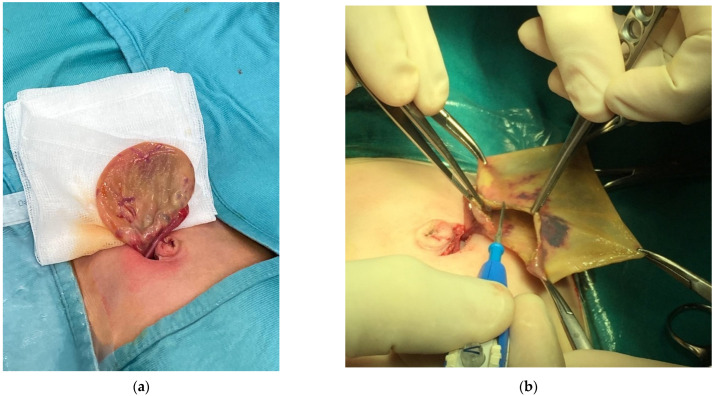
Neonatal ovarian cyst extracted to the navel (**a**) with ovarian cystectomy with salvage of a part of ovarian parenchyma (**b**).

**Figure 5 children-09-01219-f005:**
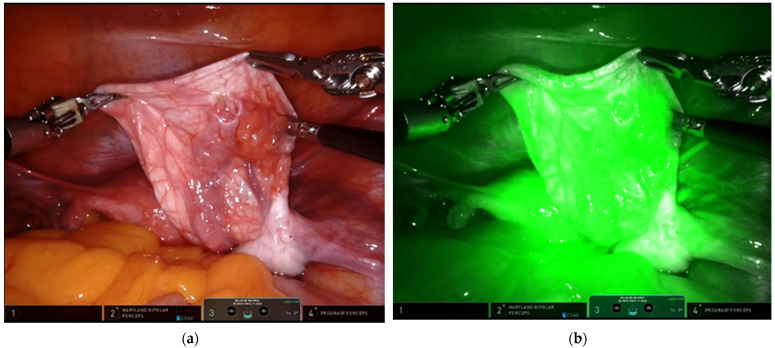
The role of ICG system to check the vascularization of the ovary after detorsion: (**a**) without ICG; (**b**) with ICG view.

**Table 1 children-09-01219-t001:** Population involved in the study.

	Total	Right Side	Laparoscopic	Robotic	ICG- Technology	Benign Lesion	Malignant Lesions
Neonates (G1)	111	73	111	0	6	111	0
Adolescents (G2)	36	18	28	8	14	30	6

**Table 2 children-09-01219-t002:** Results.

	Newborns	Teenagers
Conversion to open surgery *	0	0
Mass excision	80	30
Positive tumor makers	-	6
- Benign Lesions		3 (10%)
- Malignant Lesions		4 (66%)
Complications after surgery	0	0
Needs of re-interventions	0	0
Mean operative time before ICG system	51.4 min	53.2 min
Mean operative time after ICG system	47.2 min	37.2 min

* Number of mini-invasive surgeries hat have been converted in open surgery.

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
