# Peer review of "Review of a 25-Year Experience in the Management of Ovarian Masses in Neonates, Children and Adolescents: From Laparoscopy to Robotics and Indocyanine Green Fluorescence Technology"

_children, 2022, doi:10.3390/children9081219_

Round 1
Reviewer 1 Report
Very interesting work on a large material. A large number of operations in children up to 6 months of age is noteworthy. What were the indications for surgery in the youngest children? What was the diameter of the operated lesions, were they symptomatic? It has not been described that ovarian cysts in this age group are often functional in nature and do not require surgery in most cases. Please, respond to this remark. Besides, I have no comments.
Author Response
We would like to thank the Editor to give us the possibility of reviewing the Ms submitted to the Children Journal. We have modified the Ms according to the reviewer’s.
- Indications for surgical removal of neonatal cysts are usually based on the size of the lesion (> 4 cm) or complicated cysts (e.g. with torsion). If the lesion is less than 4 cm we follow-up because in most cases surgery is not necessary.
- The median diameter of the lesions was 5,4 cm. In adults 13 cases had ovarian torsion and
- We add in the text that ovarin cyst are often functional in nature and do not require surgery (line 53).
Reviewer 2 Report
Dear Authors
I have some questions and comments.
1. line 73 - is a small mistake - a 25-yearretrospective analysis
2. what is the explanation of the fact that the right side is more common in children?
3. line 117 - positive tumors markers group is too small group of patients for statistic analyses, it is necessarry to improve.
4. what kind of tumors markers were detected?
5. I don't agree with the conclusions - what kind of the surgery should be performed when we have malignant tumor of the ovary
Author Response
We would like to thank the Editor to give us the possibility of reviewing the Ms submitted to the Children Journal. We have modified the Ms according to the reviewer’s:
- We have correct the mistake (line 73).
- In our cases the right side is more involved, We don't know why but it would be interesting to have a study with further cases to see if there is any correlation.
- We agree. It’s a preliminary date, we need further cases about this.
- Tumor markers that were analyzed are CA125, CEA, CA 19-9. We add this information in the test (line 85).
- Mini-invasive approach is always indicated. In case of high suspicion of malignancy you have to remove mass/ovary. ICG can be useful to check lynpho-node status and to have a correct decision-making.
Round 2
Reviewer 2 Report
Dear Authors
1. I don't see information about tumors markers in the text, line 85. This sentence is deleted. Tumors markers are important and helpfull in malignant tumors diagnosis. It is necessary to establish did the authors analyse the tumor markers or no. If they don't it is a basic limitation of the study and it should be included in the text.
2. I am not sure - didn't you check other tumors markers as AFP, BHCG and LDH?
3. It is necessary to explain the abbreviation ICG fluorescence technology, not every physician knows what it means.
Author Response
We have modified the Ms according to the reviewer’s with coloured text.
- Tumor markers were analyzed preoperatively in all teenagers. In line 91-92 we wrote the markers that were made.
- We check CA125, CA19-9, CEA, AFP and B-hcg.
- As you suggest we explain ICG in the introduction (line 73).